# External Quality Assessment of Bacterial Identification and Antimicrobial Susceptibility Testing in African National Public Health Laboratories, 2011–2016

**DOI:** 10.3390/tropicalmed4040144

**Published:** 2019-12-13

**Authors:** Olga Perovic, Ali A. Yahaya, Crystal Viljoen, Jean-Bosco Ndihokubwayo, Marshagne Smith, Sheick O. Coulibaly, Linda De Gouveia, Christopher J. Oxenford, Sebastien Cognat, Husna Ismail, John Frean

**Affiliations:** 1National Institute for Communicable Diseases, Division of National Health Laboratory Service, Johannesburg 2131, South Africa; 2Department of Clinical Microbiology, University of Witwatersrand, Johannesburg 2193, South Africa; 3World Health Organization, Regional Office for Africa, Brazzaville 06, Congo; aliahmedy@who.int (A.A.Y.); ndihokubwayoj@who.int (J.-B.N.); coulibalysh@who.int (S.O.C.); 4World Health Organization, 69007 Lyon, France; oxenfordc@who.int (C.J.O.); cognats@who.int (S.C.); 5WITS Research Institute for Malaria, University of Witwatersrand, Johannesburg 2193, South Africa

**Keywords:** external quality assessment, antimicrobial susceptibility testing, national public health laboratories, African region

## Abstract

Background: In 2002, the World Health Organization (WHO) launched a regional microbiology external quality assessment (EQA) programme for national public health laboratories in the African region, initially targeting priority epidemic-prone bacterial diseases, and later including other common bacterial pathogens. Objectives: The aim of this study was to analyse the efficacy of an EQA programme as a laboratory quality system evaluation tool. Methods: We analysed the proficiency of laboratories’ performance of bacterial identification and antimicrobial susceptibility testing (AST) for the period 2011–2016. The National Institute for Communicable Diseases of South Africa provided technical coordination following an agreement with WHO, and supplied EQA samples of selected bacterial organisms for microscopy (Gram stain), identification, and antimicrobial susceptibility testing (AST). National public health laboratories, as well as laboratories involved in the Invasive Bacterial Diseases Surveillance Network, were enrolled by the WHO Regional Office for Africa to participate in the EQA programme. We analysed participants’ results of 41 surveys, which included the following organisms sent as challenges: *Streptococcus pneumonia, Haemophilus influenzae, Neisseria meningitidis, Salmonella* Typhi, *Salmonella* Enteritidis, *Shigella flexneri, Staphylococcus aureus, Streptococcus agalactiae, Streptococcus anginosus, Enterococcus faecium, Serratia marcescens, Acinetobacter baumannii*, and *Enterobacter cloacae*. Results: Eighty-one laboratories from 45 countries participated. Overall, 76% of participants obtained acceptable scores for identification, but a substantial proportion of AST scores were not in the acceptable range. Of 663 assessed AST responses, only 42% had acceptable scores. Conclusion: In the African Region, implementation of diagnostic stewardship in clinical bacteriology is generally suboptimal. This report illustrates that AST is poorly done compared to microscopy and identification. It is critically important to make the case for implementation of quality assurance in AST, as it is the cornerstone of antimicrobial resistance surveillance reporting and implementation of the Global Antimicrobial Resistance Surveillance System.

## 1. Introduction

Antibiotic resistance is a worldwide health threat: at least 700,000 people die annually due to antibiotic-resistant pathogens, and this figure is projected to reach 10 million by 2050 [1]. It is a severe and growing global health security risk, which needs to be prioritised at country, regional and international levels.

The development and implementation of national antimicrobial resistance (AMR) plans of action that complement international efforts are a major step towards containment. Global partnerships need to be strengthened, because the responsibility for reducing AMR calls for collaborative action in all sectors (human health, animal health, agriculture and environment), in line with the ‘One Health’ approach.

Globally, a projected 50% of antibiotic agents can be acquired without a prescription, predominantly on the informal market in some parts of Africa, South America and Asia [1]. In the World Health Organization (WHO) African Region, many countries have developed regulations that restrict the access to antibiotics without a prescription; however, these regulations are not enforced in three-quarters of African Region nations [2]. This problem is not confined to developing countries, however. In 2014, a WHO report showed that 19 European countries had unrestricted access to antibiotics [2,3], which supports evidence for the overuse of antibiotics and the increasing threat of resistant pathogens.

The detection and monitoring of the spread of AMR rests on laboratory testing for phenotypic and genetic markers of resistance. Laboratory-based surveillance relies on accurate antimicrobial susceptibility testing (AST) data. Additionally, specimen submission practices and collection of demographic, clinical and other data using adequate protocols all contribute to the establishment of epidemiologically-driven surveillance programmes.

For credible and reliable AMR surveillance, it is essential that countries support laboratory systems capable of providing good quality data in a standardised format.

In the WHO African Region, implementation of diagnostic quality assurance in clinical bacteriology is lacking in several national reference laboratories that are on the front line for confirmation of priority outbreak-prone diseases.

In July 2002, WHO launched a regional microbiology external quality assessment (EQA) programme for national public health and other laboratories in the African Region, initially targeting priority epidemic-prone bacterial diseases, and later including other bacterial pathogens [4], predominantly common causes of meningitis, diarrhoea, and sepsis. The National Institute for Communicable Diseases (NICD), a division of the National Health Laboratory Service of South Africa, provided technical coordination following agreements with WHO (Lyon office and later, the Regional Office for Africa) and advised WHO on participating laboratories’ requirements to correct deficiencies and maintain proficiency. Consequently, NICD commenced provision of EQA samples for microscopy (Gram stain), identification, and AST of selected bacterial organisms.

For this report, we analysed the performance of participating laboratories over the last six years of the programme, from 2011 to 2016. The first ten years of the programme have been reported previously [5].

## 2. Methods

### 2.1. Study Design

From each participating country (Figure 1), a national public health laboratory recommended by its Ministry of Health, as well as laboratories involved in the Invasive Bacterial Diseases Surveillance Network (some functioned as both), were enrolled by the WHO Regional Office for Africa [4,5]. Each laboratory received a batch of samples of bacterial isolates and simulated clinical sample slides for Gram stain (for the meningitis pathogens), prepared by NICD, three times per year. Based on bacterial species and whether the isolate was lyophilized or a simulated sample, preparation time ranged from one month to one week before the shipment. Retained survey samples were cultured weekly for quality control, homogeneity, and stability until survey was closed. Simultaneously, a questionnaire was sent to participating laboratories for continuous capacity assessment, for example to obtain information on their facilities, capacity, and staff competence. Preparation of the samples included quality control and validation of survey material before dispatch. Concurrently, five referee laboratories (three from South Africa and two from the United States) tested all the isolates and their results determined whether the specimen panels were suitable for evaluation of participants’ responses.

For this analysis, participating laboratories’ bacterial identification and AST results for the period 2011–2016 were identified from the programme database.

The programme mirrored the Clinical Microbiology Proficiency Testing (CMPT) programme from the Department of Pathology and Laboratory Medicine, University of British Columbia, Vancouver, Canada [6]. The content of the panels of specimens to be distributed in the programme was decided by a Technical Implementation Group at NICD and was reviewed at the programme’s Regional Advisory Group (RAG) annual meetings. RAG members were recruited from NICD, employees from AFRO WHO and WHO Lyon offices. Antibiotic panels were selected based on Clinical and Laboratory Standards Institute (CLSI) or European Committee on Antimicrobial Susceptibility Testing (EUCAST) [7,8] guidelines for first-line treatment options for each organism; for example, the following antibiotics were selected for *Streptococcus pneumoniae*: penicillin or oxacillin, erythromycin, ceftriaxone or cefotaxime, and trimethoprim-sulfamethoxazole [7,8]. Additional relevant laboratory procedures were also assessed, such as tests for beta-lactamases. In each survey, AST evaluation of at least two organisms was required; some organisms were sent only once during the period under review. Design of the surveys was done annually, based initially on regional priority organisms causing epidemic diseases, and later covered common general bacterial pathogens. Our program did not include anaerobic organisms in consideration of priorities of participants. 

The grading scheme for bacterial species identification assessed responses based on clinical impact on an imaginary patient and was approved by the Technical Implementation Group and the RAG. The most correct answer was awarded a score of 4/4; an answer that was not completely correct, with no or little clinical impact, scored 3/4; a score of 1/4 was awarded for an answer that could lead to a minor diagnosis or treatment error, and a zero score was assigned for a major diagnosis or treatment error. If responses to the identification of organisms from culture were incorrect, the AST component was automatically scored as 0. The 2/4 score was omitted to emphasise the difference between acceptable and unacceptable performance [6].

The scoring of the AST was based on the qualitative criteria for susceptible, intermediate and resistant determination:Correct (scored 4/4)–the participant results agreed with those from the referee laboratories.Minor error (scored 3/4)–the participant results indicated an ‘intermediate’ susceptibility result, and the referee laboratories reported either a susceptible or a resistant result.Major error (scored 1/4)–the participant results indicated the microorganism as resistant to the antimicrobial, whereas the referee laboratories reported the isolate as susceptible.Very major error (scored 0/4)–the most serious error; the participant results indicated that the microorganism was susceptible to the antimicrobial, whereas the referee laboratories indicated that the microorganism was resistant [9].

Laboratory performance is presented as the proportion (%) of participants that achieved scores of 3 or 4 (acceptable performance) for pathogen identification and AST. Not all participants returned responses to all surveys.

The EQA programme followed the ISO/IEC 17043 guidelines [5] and all samples were packaged and shipped according to International Air Transport Association Packing Instructions P650 [5].

Consistent non-responders were alerted at least three times and if they still did not respond, they were removed from the pool of participants. The programme was reviewed and participants added or removed at the annual RAG meetings.

### 2.2. Data Analysis

Responses and scores were recorded on a customised Microsoft Access database (Microsoft, Redmond, Washington, United States) and analysed using Microsoft Excel (Microsoft, Redmond, Washington, United States). Proportions for acceptable (scores 3/4 or 4/4) and non-acceptable (0/4 or 1/4) responses were compared using the Pearson’s Chi-square test. The level of significance was set at *p* < 0.05.

### 2.3. Ethical Considerations

As no human or animal material was used for this programme and the external quality assessment programme followed ISO 17043 guidelines, no ethical permission was required. Based on ISO 17043 guidelines, individual participants’ data were confidential and protected.

## 3. Results

From questionnaire feedback, the majority of responding laboratories (53/61, 87%) used basic biochemical tests (catalase, oxidase, indole, etc.) for phenotypic identification of organisms and Kirby–Bauer disk diffusion method for AST. Two laboratories performed minimal inhibitory concentration (MIC) assays using gradient concentration strips such as E-test (bioMérieux, Marcy-l’Étoile, France) for *S. pneumoniae*. Seven laboratories used Vitek II (bioMérieux, Marcy-l’Étoile, France), and one used a MicroScan WalkAway system (Siemens Healthcare Diagnostics, Los Angeles, CA, United States) for identification and AST. Laboratories interpreted results and reported categorical susceptibility, based on CLSI or EUCAST guidelines. (However, no participating laboratory used the latest CLSI guideline).

The total number of participants was 81 from 45 countries (Figure 1). For organism identification (listed in the Table 1), 1662/2184 (76%) of responses over 41 surveys were scored as acceptable (Table 1). One third of participants were non-responders.

AST for *S. pneumoniae* showed no improvement in performance over the period (Figure 2). The pathogen with the lowest number of acceptable AST results (average of 48.5% of participants achieved acceptable scores) was *Haemophilus influenzae* (Figure 2). The specific test for detection of beta-lactamase in this organism showed no significant variation in proportion of acceptable results (*p* = 0.5). 

For enteric pathogens, AST results from an average of 48 responding laboratories showed a significant decrease in the percentage of laboratories reporting acceptable AST results for *Salmonella* Typhi (*p* < 0.005) and a significant increase in acceptable performance for *Salmonella* Enteritidis and *Shigella flexneri*, (*p* < 0.005) (Figure 3).

*Staphylococcus aureus* acceptable AST results from 57 participants showed a significant decline over a period of 3 years (*p* < 0.005) (Figure 4) due to the lack of appropriate interpretation of clindamycin-inducible resistance, whereas *Pseudomonas aeruginosa* susceptibility testing showed improvement. AST for *Streptococcus anginosus* presented the biggest challenge, with only 29% of responding laboratories reporting acceptable results (Figure 5).

## 4. Discussion

Most of the participating laboratories in this EQA programme were from 28 low-income countries in Africa, according to the World Bank classification. These laboratories face many challenges, ranging from maintaining basic infrastructure (such as electricity and water supply), to limited funds for consistent supply of basic essential laboratory consumables, backup solutions such as uninterruptible power supplies or generators, and internet connections. Only a few laboratories have automated instruments for bacterial identification and AST. While the majority of laboratories were not accredited to any national or international standard, at least half of the participants were in the early stages of the WHO Stepwise Laboratory Quality Improvement Process Towards Accreditation programme [10].

Problems leading to incorrect results were numerous; for example, misidentification of the pathogen, which automatically resulted in a score of 0 for AST, as use of antibiotics is organism dependent. Use of incorrect media for AST (e.g., chocolate agar used for *H. influenzae* instead of CLSI- or EUCAST-recommended media), incorrect testing methodology, and incorrect interpretation of results were other reasons for unacceptable performance. While for many routine laboratories, full AST for *H. influenzae* is unnecessary, the participants represent national public health reference diagnostic facilities, and therefore could reasonably be expected to have the technical capacity for *H. influenzae* AST.

Based on information from the questionnaires obtained from participants, the most challenging problem was media production, as the majority of the laboratories produce their own media using human blood, instead of the required horse or sheep blood. Further, very poor internal quality control procedures were indicated, with almost no laboratories using the strains recommended by appropriate AST guidelines for quality control. There were participants with no access to the latest standards for the interpretation of results, although they had been referred to the freely-available EUCAST method in the programme commentary. Sometimes outdated versions of AST guidelines were used, and some laboratories used EUCAST for interpretation but did not follow the methodology.

Identification of bacterial pathogens was generally less problematic than AST, as reported previously [5]. Antimicrobial susceptibility testing remains a huge challenge, as the majority of laboratories did not consistently achieve acceptable AST results, based on the program criteria [6].

Laboratory performance for identification of enterococci and streptococci, and for AST, was very poor. Possible reasons for this include the use of inappropriate, expired or poorly quality-controlled media and reagents, including antibiotic discs, and a lack of incubation temperature control. Based on ongoing poor performance for identification and AST, we recorded a lack of quality management systems in the laboratories for dealing with non-conformance. This suggests a deficiency in laboratory leadership to implement regular corrective actions, as reflected by laboratories repeating the same mistakes across surveys. It is important to note that laboratories rarely have support from clinical microbiologists, based on responses received from our evaluation forms.

One-third of participants were persistent non-responders, which is consistent with the previous study described by our group [5]. Following RAG meeting discussions, some laboratories were removed from the programme after three or more consecutive non-responses. This programme provided an opportunity to assess and improve microbiological analysis and reporting of results, and should have been utilised as a channel for continuous training of existing and new laboratory personnel. RAG meetings were convened annually to discuss laboratories’ performance, and identify critical gaps; these also led to recommendations for future programme activities. Unfortunately, very few laboratories showed any improvement and only five laboratories achieved aggregated acceptable results over the period reviewed.

There is a need to build clinical laboratory capacity, as the majority of communicable disease surveillance programmes are laboratory based [11]. Strategies for establishing, strengthening, guaranteeing and maintaining the quality of laboratory test results are critical for surveillance initiatives. All diagnostic facilities should have procedures for ongoing assessment of the quality of test reagents and test performance. In addition to implementing internal quality control practices, laboratories should also participate in EQA programmes. Building clinical laboratory capacity will enable adequate and reliable data that can guide policy actions to combat AMR.

Implementation of a sustainable quality management system that qualifies for a recognised accreditation scheme would be the optimal solution and would provide confidence in the quality of laboratory results. Laboratory assessments should emphasise implementation of diagnostic stewardship programmes, which improve the utilisation of microbiological tests and results at the clinician level.

In a stepwise approach for implementation of AMR surveillance in Africa [11,12,13,14], the emphasis is on the development of laboratory capability to perform surveillance activities. To apply these activities, commitment is essential, using the opportunities provided by existing international frameworks, such as the International Health Regulations (2005) [15], the WHO Global Action Plan on AMR [16] and the One Health approach. In addition, WHO collaborating centre networks for AMR may facilitate these activities.

Such commitments include development of policies, plans and budgets [11]. Each country should start with a gap analysis to understand the level of their laboratory capabilities and build on the existing platform.

The Freetown Declaration was launched in 2015 to establish public health laboratory networks for AMR surveillance in Africa, and AMR was also highlighted in the regional strategy for health security and emergencies adopted by member states during the regional WHO committee meeting in 2016 [13,14].

## 5. Conclusions

This report illustrates that AST is poorly performed in Africa. It is critically important to make the case for building sustainable capacity for bacteriology, including AST, in most of the national public health laboratories in Africa. Along with the implementation of a robust quality management system for AST, this will formulate the AMR surveillance reporting and implementation of the Global Antimicrobial Resistance Surveillance System more appropriately [13].

Primarily, laboratory quality assessment and gap analysis should be the foundation for developing AMR surveillance for global reporting.

This report indicates an urgent need for clinical bacteriology laboratory capacity-building on the continent, as AMR was only recently recognised as a critical problem. 

## Figures and Tables

**Figure 1 tropicalmed-04-00144-f001:**
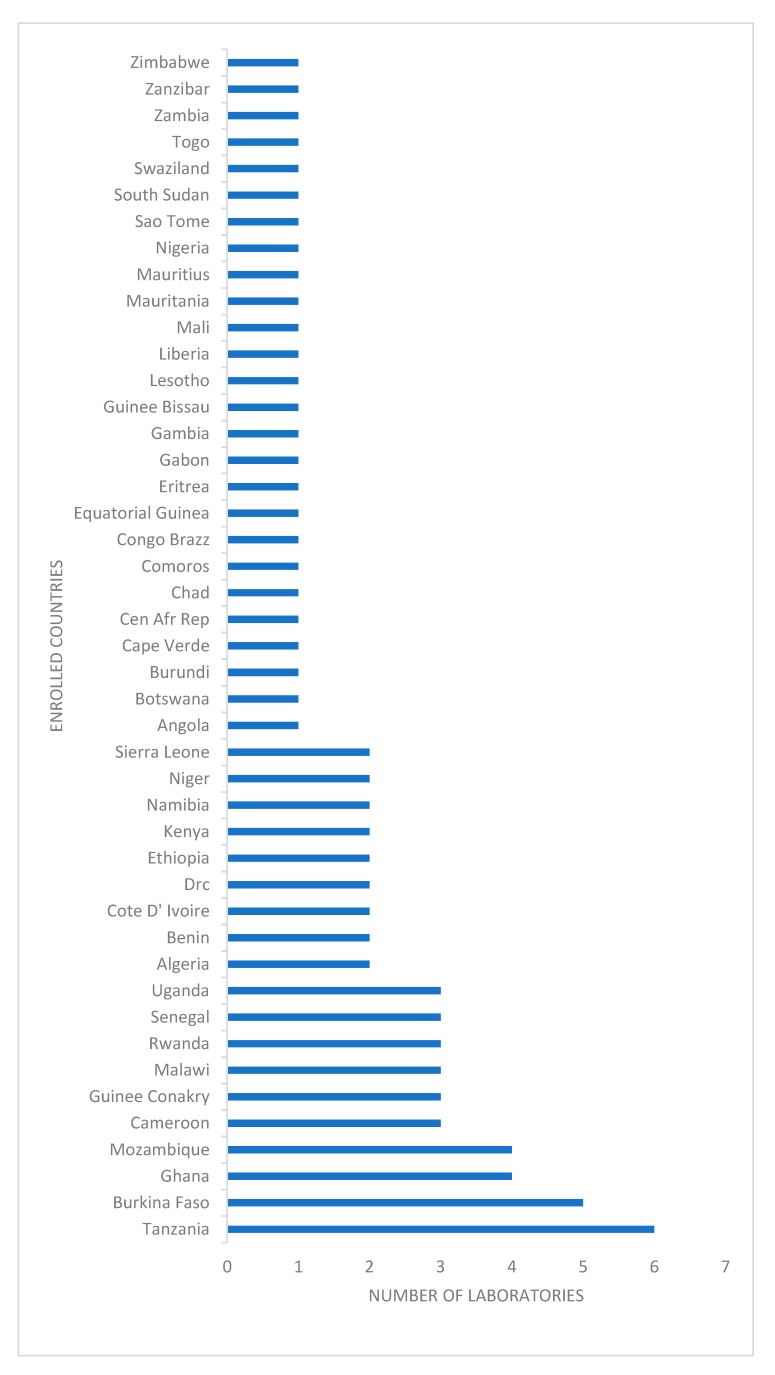
Enrolled countries in the programme with number of participating laboratories, World Health Organization and National Institute for Communicable Diseases external quality assessment programme for national public health laboratories, 2011–2016.

**Figure 2 tropicalmed-04-00144-f002:**
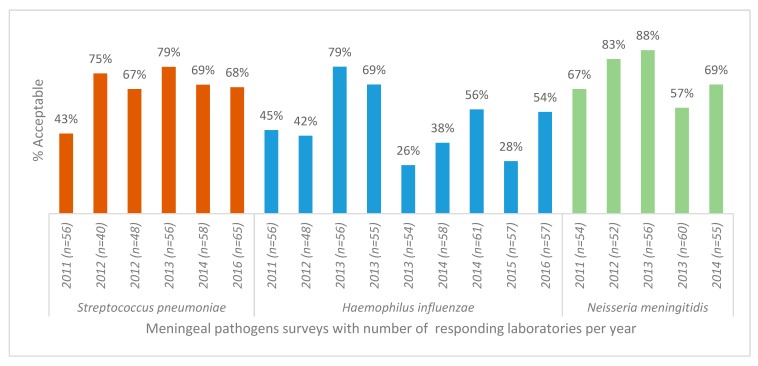
Antimicrobial susceptibility testing results for *S. pneumoniae, H. influenzae*, *N. meningitidis* isolates, World Health Organization and National Institute for Communicable Diseases external quality assessment programme for national public health laboratories, 2011–2016 (*n* = number of responding laboratories).

**Figure 3 tropicalmed-04-00144-f003:**
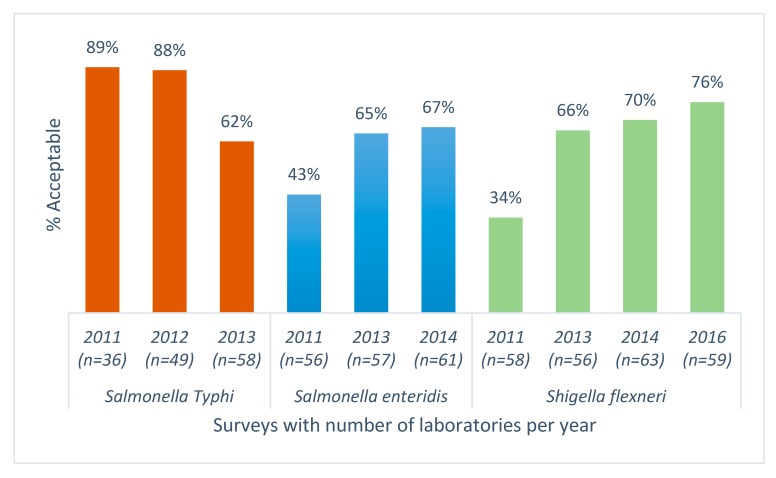
Antimicrobial susceptibility testing results for *Salmonella* Typhi, *Salmonella* Enteritidis, *Shigella flexneri*; World Health Organization and National Institute for Communicable Diseases external quality assessment programme for national public health laboratories, 2011–2016 (*n* = number of responding laboratories).

**Figure 4 tropicalmed-04-00144-f004:**
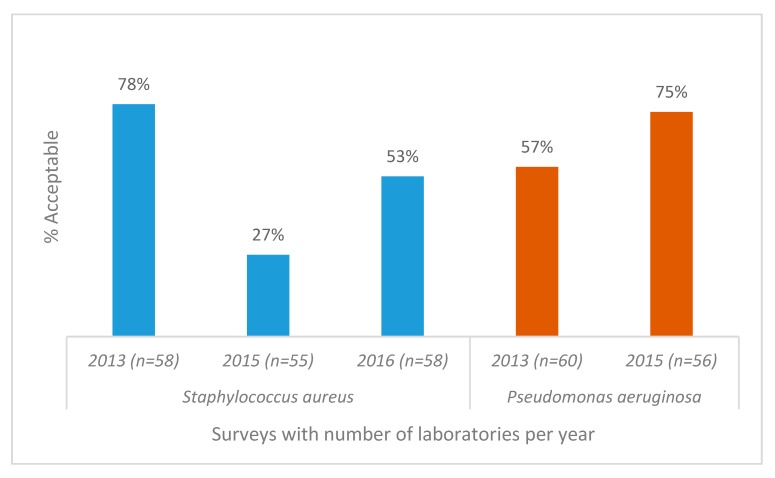
Antimicrobial susceptibility testing results for *Staphylococcus aureus* and *Pseudomonas aeruginosa*, World Health Organization and National Institute for Communicable Diseases external quality assessment programme for national public health laboratories, 2011–2016 (*n* = number of responding laboratories). The poor performance in 2015 was due to misinterpretation of the clindamycin D zone for *S. aureus.*

**Figure 5 tropicalmed-04-00144-f005:**
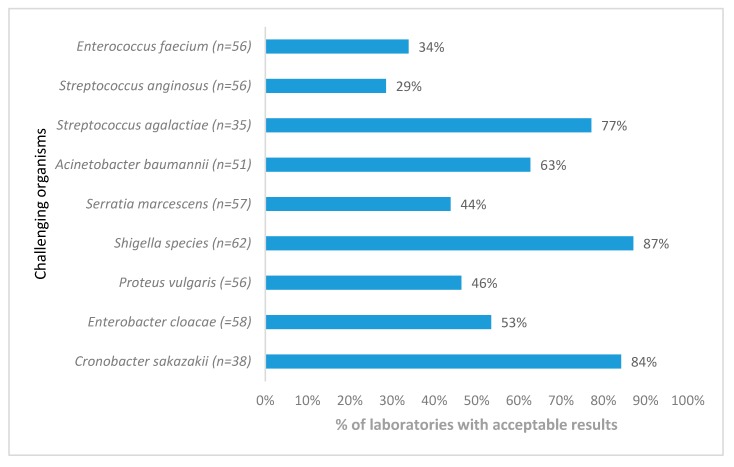
Antimicrobial susceptibility testing performance for Gram-positive and Gram-negative organisms only distributed once, World Health Organization, National Institute for Communicable Diseases external quality assessment programme for national public laboratories, 2011–2016 (n = number of responding laboratories). AST, antimicrobial susceptibility testing.

**Table 1 tropicalmed-04-00144-t001:** Summary of performance in identification of bacteria, World Health Organization and National Institute for Communicable Diseases external quality assessment programme for national health laboratories, 2011–2016.

Organisms	Number of Surveys	Average Number of Participating Laboratories per Survey	Average Number (%) with Acceptable Identification
**Meningeal pathogens**			
***Streptococcus pneumoniae***	7	54	46 (83)
***Haemophilus influenzae***	9	56	38 (67)
***Neisseria meningitidis***	6	55	51 (86)
**Enteric pathogens**			
***Salmonella* Typhi**	3	48	36 (75)
***Salmonella* Enteritidis**	3	58	30 (72)
***Shigella flexneri***	4	59	37 (84)
**Other Gram-positive pathogens**			
***Staphylococcus aureus***	3	57	51 (89)
***Streptococcus agalactiae***	1	37	33 (89)
***Streptococcus anginosus***	1	56	36 (64)
***Enterococcus faecium***	1	56	21 (37)
**Other Gram-negative pathogens**			
***Serratia marcescens***	1	57	31 (54)
***Acinetobacter baumannii***	1	51	32 (62)
***Enterobacter cloacae***	1	58	40 (69)

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
