# Peer review of "External Quality Assessment of Bacterial Identification and Antimicrobial Susceptibility Testing in African National Public Health Laboratories, 2011–2016"

_tropicalmed, 2019, doi:10.3390/tropicalmed4040144_

Round 1

Reviewer 1 Report

This is an important study that highlights suboptimal antimicrobial susceptibility testing in low income countries in Africa. The associated manuscript is well-written i.e. the findings are clear and succinct. My comments are all minor.

Could add some detail in the methods

There were 5 referee laboratories; were each of the panels reviewed by all 5 prior to distribution or was only a single lab charged with production of a given panel? Where were the referee labs located? (if remote/outside Africa, could transport/logistics have impacted results?)What was the time from preparation to testing? Selection: was this the complete list of countries and laboratories that were approached. May want to be explicit about the selection/participation in the methods. Is the grading scheme for species identification standard? Odd question but why does the grading not include 2/4? Could elaborate (briefly) why how the panel of organisms was selected (seems pretty obvious but could use a few words to explain that these are most frequently implicated in enteric disease/meningitis Why were anaerobic organisms not included?

Author Response

To editor

Please find authors responses:

Reviewer: Could add some detail in the methods

1.1 There were 5 referee laboratories; were each of the panels reviewed by all 5 prior to distribution or was only a single lab charged with production of a given panel? Where were the referee labs located? (if remote/outside Africa, could transport/logistics have impacted results?)

My response - Text amended to: Concurrently, five referee laboratories (three from South Africa and two from the United States) tested all the isolates and their results determined whether the specimen panels were suitable for evaluation of participants’ responses.

(Because of the quality control measures described in the next point (below), the effect of delays in transportation were able to be detected).

1.2 What was the time from preparation to testing?

My response – text amended to: Based on bacterial species and whether the isolate was lyophilized or a simulated sample, preparation time ranged from one month to one week before the shipment. Retained survey samples were cultured weekly for quality control, homogeneity, and stability until survey was closed.

1.3 Selection: was this the complete list of countries and laboratories that were approached.

My response: Yes, it is the complete number of countries approached and enrolled.

1.4 May want to be explicit about the selection/participation in the methods.

My response – text amended to: From each participating country (Figure 1), a national public health laboratory recommended by its Ministry of Health, as well as laboratories involved in the Invasive Bacterial Diseases Surveillance Network (some functioned as both), were enrolled by the WHO Regional Office for Africa.

1.5 Is the grading scheme for species identification standard?

My response – text amended to: The grading scheme for bacterial species identification assessed responses based on clinical impact on an imaginary patient and was approved by the Technical Implementation Group and the RAG.

1.6 Odd question but why does the grading not include 2/4?

My response – text amended to: The 2/4 score was omitted to emphasise the difference between acceptable and unacceptable performance.6

1.7 Could elaborate (briefly) why how the panel of organisms was selected (seems pretty obvious but could use a few words to explain that these are most frequently implicated in enteric disease/meningitis Why were anaerobic organisms not included? 

My response - text amended to:  Design of the surveys was done annually, based initially on regional priority organisms causing epidemic diseases, and later covered common general bacterial pathogens. Our program did not include anaerobic organisms in consideration of priorities of participants.

Reviewer 2 Report

Article written clearly. In my opinion does not require corrections . A great value of the work is a discussion explaining the reasons for insufficient, incorrect diagnosis of infections in laboratories, which can help improve the conditions and work of microbiologists in this particular part of the world.

Author Response

Nothing to report 

Reviewer 3 Report

This report is aimed to analyse the efficacy of a regional microbiology external quality assessment programme as a laboratory quality system evaluation tool. This report indicates that the antimicrobial susceptibility testing is poorly done compared to microscopy and identification. This report is well-written and easy to follow. The data provided and discussed in this report are useful to help the clinical laboratories in African to implement of quality assurance in antimicrobial susceptibility testings.

Minor points:

1. Line 167-170.

The format in table 1 should be adjusted to make the descriptions more clear. For example, the spellings of some vocabularies are divided into two lines. Otherwise, in general, the “vertical line” is seldom used in table.

2. line 171-174:

As for the information showed by the figure 1, “the descriptions in text” or a table may be more suitable to present the participating laboratories in particular countries.

Author Response

Reviewer 2. Minor points:

Line 167-170.

The format in table 1 should be adjusted to make the descriptions more clear. For example, the spellings of some vocabularies are divided into two lines. Otherwise, in general, the “vertical line” is seldom used in table.

My response: Table reformatted and vertical lines removed.

line 171-174:

As for the information showed by the figure 1, “the descriptions in text” or a table may be more suitable to present the participating laboratories in particular countrie.

My response: I prefer the graph as it gives a more immediate visual impression.